# Short-Term Effects of Different Forest Management Methods on Soil Microbial Communities of a Natural *Quercus aliena* var. *acuteserrata* Forest in Xiaolongshan, China

**Pan Wan [1], Gongqiao Zhang [1], Zhonghua Zhao [1], Yanbo Hu [1], Wenzhen Liu [2] and Gangying Hui [1,\*]**

[1] Key Laboratory of Tree Breeding and Cultivation of State Forestry Administration, Chinese Academy of Forestry, Beijing 100091, China; wp7413841@163.com (P.W.); zhanggongqiao@126.com (G.Z.); zzhwl19780101@163.com (Z.Z.); hyanbo@caf.ac.cn (Y.H.)

[2] Xiaolongshan Research Institute of Forestry of Gansu Province, Tianshui 741000, China; Liu_wenzhen@163.com

[*] Correspondence: hui@caf.ac.cn; Tel.: +86-010-6288-8854

**Abstract:** One of the aims of sustainable forest management is to preserve the diversity and resilience of ecosystems. Unfortunately, changes in the soil microbial communities after forest management remain unclear. We analyzed and compared the soil microbial community of a natural *Quercus aliena* var. *acuteserrata* forest after four years of four different management methods using high-throughput sequencing technology. The forest management methods were close-to-nature management (CNFM), structure-based forest management (SBFM), secondary forest comprehensive silviculture (SFCS) and unmanaged control (CK). The results showed that: (1) the soil microbial community diversity indices were not significantly different among the different management methods. (2) The relative abundance of *Proteobacteria* in the SBFM treatment was lower than in the CK treatment, while the relative abundance of *Acidobacteria* in the SBFM was significantly higher than that in the CK treatment. The relative abundance of *Ascomycota* was highest in the CNFM treatment, and that of *Basidiomycota* was lowest in the CNFM treatment. However, the relative abundance of dominant bacterial and fungal phyla was not significantly different in CK and SFCS. (3) Redundancy analysis (RDA) showed that the soil organic matter (SOM), total nitrogen (TN), and available nitrogen (AN) significantly correlated with the bacterial communities, and the available potassium (AK) was the only soil nutrient, which significantly correlated with the composition of the fungal communities. The short-term SBFM treatment altered microbial bacterial community compositions, which may be attributed to the phyla present (e.g., *Proteobacteria* and *Acidobacteria*), and the short-term CNFM treatment altered microbial fungal community compositions, which may be attributed to the phyla present (e.g., *Ascomycota* and *Basidiomycota*). Furthermore, soil nutrients could affect the dominant soil microbial communities, and its influence was greater on the bacterial community than on the fungal community.

**Keywords:** different forest management methods; soil microorganisms; soil nutrients; *Quercus aliena* var. *acuteserrata*

## 1. Introduction

The soil microbial community is a vital component of the soil biological system. It plays important roles in the nutrient cycles and energy flows, providing essential services to the forest ecosystem [1–4]. Bacteria and fungi are present in high proportions in soil microbial communities [5]. Among them,

soil fungi have the function of catalyzing the turnover of complex organic resources, which can drive the degradation of organic matter macromolecules [6–8]. Bacteria generally utilize the easily available substrates decomposed by fungi. Moreover, it can facilitate fungal degraders by providing electrons or essential micronutrients [9,10]. In forest soils, microbial communities are affected by the changes in aboveground vegetation communities and soil environmental properties (such as nutrients, temperature, and moisture) influenced by human activity through forest management [11,12]. Generally, the changes in soil properties are directly affected by litter, roots, and plant exudates of aboveground vegetation, and then influence the soil microbial communities [13–15]. Plants can selectively attract and maintain rhizosphere microbes by root exudates, and at the same time, the microbial may strongly affect the growth of plants by releasing mineral elements [16,17]. Thus, the interaction between aboveground vegetation and soil microbial communities can influence the process of the forest ecosystem [18,19]. Hence, it is necessary to understand the relationships between aboveground vegetation, soil properties, and soil microbial communities in forest ecosystems.

Forest management can affect multiple properties of soil (such as pH, bulk density, organic matter, soil structure, and soil microclimate) due to changes in the dominant tree species, canopy densities, and forest microclimate, which may indirectly influence the soil microbial community [20]. Forest management has been shown to influence microbial community structure and composition through changes in aboveground species composition or with type of harvesting strategies [11,12,21]. Despite the extensive research on the effect of forest management on soil microbial community, most of these studies are focused on the intensity of the management intervention while there is little research on the effect of different management methods. Previous studies found that soil organic carbon and total nitrogen (TN) are significantly different between tree harvest methods. For example, the sawlog harvesting and whole-tree harvesting both cause increases in soil carbon and nitrogen [22]; forest harvest practices that remove more than the tree bole significantly reduce soil TN, microbial biomass carbon and nitrogen, and the persistent reduction of soil TN and microbial biomass in the severe harvest treatments [23]. Hence, we hypothesize that forest management may result in the alteration of soil nutrients and microbial community; the hypotheses are: (1) the forest management did not change the levels of soil nutrients in the short term; (2) forest management measures had altered soil microbial community compositions, and these changes are different in different forest management methods; (3) soil nutrients could affect the dominant soil microbial communities. Therefore, understanding the changes in soil microbial communities after different forest management methods will be undoubtedly necessary for forest management. Currently, more methods are used in soil microbial community studies, such as cloning libraries, PCR-denaturing gradient gel electrophoresis (DGGE,) and phospholipid fatty acid (PLFA) [24,25]. Compared with these methods, the high-throughput sequencing technology has the characteristics of obtaining large amounts of data, short sequencing cycles, high rates of accuracy, and low cost. Therefore, the use of this method can provide obvious advantages for understanding the microbial ecology in forest ecosystems [26–30].

A mixed *Quercus aliena* var. *acuteserrata* forest is one of the most important forest types in China, and has ecological functions such as water and soil conservation, improvement of soil fertility and stress resistance [31]. Close-to-nature forest management (CNFM), structure-based forest management (SBFM), and secondary forest comprehensive silviculture (SFCS) are three forest management methods that have been widely applied in China. SBFM relies on stand spatial structure parameters to optimize the spatial structure of the forest, which aims to cultivate healthy, stable, and high-quality forests [32,33]. CNFM focuses on ecological and environmental feasibility, and it aims to cultivate forests with high stability and long-term productivity [34,35]. SFCS concerns adjusting the stand density, and its management purpose is to achieve sustainable timber usage [36]. The three forest management methods have been applied for improving the quality of mixed oak forests. Observations of the impacts of different forest management methods on natural forest of mixed oak have primarily concentrated on the growth, structure of forest, and soil nutrient cycling (carbon and nitrogen) [37–40].

There are comparatively few studies about the accompanying changes in soil microbial communities of forest after tending in this area of China. Changes in soil microbial communities with the three forest management methods are unclear. Getting this information is critical for understanding the fundamental ecological processes after tending with different management methods.

Hence, we analyzed and compared the characteristics in soil nutrients and the microbial community in the natural mixed forest of *Quercus acuminatus* with four management method treatments after four years using high-throughput sequencing technology in this study. The objectives of this study were to: (1) analyze and compare the changes in soil nutrients, microbial community compositions, and diversity, and (2) explore the effects of soil nutrients on the soil microbial community. It is expected that the results of the study would be useful for monitoring the soil ecosystem development of mixed oak forest with different management methods.

## 2. Materials and Methods

### 2.1. Study Area Description

The research area was Baihua Forest Farm in the Xiaolongshan Nature Reserve, which is located in Daganzigou, Gansu Province, China. The forest is located in the Western Qinling Mountains, with geographical coordinates of 34°16′-35°25′ N, 106°15′–106°30′ E and an altitude of 1400–2500 m. The area is in the transition zone between a warm temperate zone to the north and a subtropical zone to the south, and thus has the climatic characteristics of both Northern and Southern China. Most of the study area is warm and humid, with a semi-humid continental monsoon climate. The annual average temperature is 7–12 °C. The extreme maximum and minimum temperatures are 39.2 and −18.2 °C, respectively. The annual precipitation is 600–900 mm and is concentrated in July, August, and September. The relative humidity in the forest area is 68%–78%, the annual evaporation is 989–1658 mm, the annual sunshine duration is 1520–2313 h, and the frost-free period is 130–220 day. The soil types in the Western and Southern Qinling Mountains are grey cinnamonic and yellow cinnamon soil, respectively, which indicates that the soils of the Xiaolongshan Mountain area have an obvious vertical distribution. The main vegetation is highly naturalized mixed pine-oak forest, with the dominant species being *Q. aliena*, alongside approximately 30 other species, such as *Quercus liaotungensis* Koidz, *Pinus armandii* Franch, and *Carya cathayensis* Sarg.

### 2.2. Soil Sampling Collection

In Spring 2013, sixteen plots (20 × 20m$^2$) were randomly established at Baihua Forest Farm. The location and site conditions are almost the same. In Autumn 2013, CNFM began in the A1–A4 plots; SBFM began in the B1–B4 plots; and SFCS began in the D1–D4 plots. No forestry operations were conducted in the C1–C4 control plots as an unmanaged control (CK) treatment. For the implementation plan of different forest management methods, see Supplementary Materials, and the differences among the three forest management methods are shown in Table S1 (Supplementary Materials). In September 2017, sixteen plots subjected to three forest management treatments and unmanaged treatments were selected for sampling. Four replicate plots were similar in slopes, slope aspects, and altitudes. To avoid edge effects, a buffer zone (5 m) was set in each plot. The stand characteristics are shown in Table 1.

Soil cores were taken to a depth of 0–20 cm, and were collected with an "S" shape, excluding the litter layer, with 4.5 cm in diameter of soil cylinders. For each plot, nine point samples were randomly chosen for soil sample collection. Then, these nine samples from each patch type were mixed to form one composite soil sample. Thus, in total, 16 composite samples were collected. The samples were stored in an ice cooler during transportation and were processed immediately upon returning to the laboratory. Each sample was portioned into two halves, one half for biological analysis and the other half for soil chemical analysis. The samples for biological analysis were stored at −80 °C; the samples for soil chemical analysis were air-dried, sieved, and stored at 4 °C until further processing.

**Table 1.** Basic characteristics of plots.

| Plot | Elevation (m) | Slope (°) | Aspect | DBH (cm) | Height (m) | Canopy Cover |
|------|---------------|-----------|--------|----------|------------|--------------|
| A1 | 1749 | 27 | East | 16.60 | 12.24 | 0.80 |
| A2 | 1727 | 34 | East | 13.10 | 12.27 | 0.80 |
| A3 | 1699 | 35 | East | 12.72 | 12.88 | 0.80 |
| A4 | 1709 | 35 | East | 12.48 | 12.70 | 0.80 |
| B1 | 1664 | 36 | East | 15.54 | 12.49 | 0.80 |
| B2 | 1727 | 37 | East | 13.37 | 12.17 | 0.80 |
| B3 | 1686 | 37 | East | 13.69 | 12.96 | 0.80 |
| B4 | 1680 | 35 | East | 15.00 | 12.73 | 0.80 |
| C1 | 1663 | 36 | East | 14.00 | 12.48 | 0.90 |
| C2 | 1717 | 35 | East | 14.42 | 11.94 | 0.90 |
| C3 | 1760 | 36 | East | 14.96 | 12.04 | 0.90 |
| C4 | 1735 | 34 | East | 14.50 | 12.88 | 0.90 |
| D1 | 1711 | 36 | East | 14.36 | 13.16 | 0.80 |
| D2 | 1700 | 36 | East | 13.91 | 12.03 | 0.80 |
| D3 | 1728 | 37 | East | 12.30 | 12.58 | 0.80 |
| D4 | 1683 | 37 | East | 13.22 | 12.24 | 0.80 |

A indicates close-to-nature forest management (CNFM), B indicates structure-based forest management (SBFM), C indicates unmanaged control (CK), and D indicates secondary forest comprehensive silviculture (SFCS).

### 2.3. Soil Sample Chemical Analysisd

Soil pH was measured with a calibrated pH meter (LEICI, Shanghai, China) (soil: water ratio of 1:2.5, *w/v*). Soil organic matter (SOM) was determined by the K2Cr2O7 colourimetric method [41]. Total nitrogen (TN) as analyzed using a Foss Kjeltec 8400 analyzer unit (Kjeltec Analyzer Unit, Foss Tecator AB; Hoganas, Sweden). Total phosphorus was measured using an ultraviolet spectrophotometer (Shimadzu Corporation, Tokyo, Japan). Available phosphorus (AP) was extracted with 0.5 mol/L NaHCO3 and measured with a spectrophotometer (Mapada Corporation, Shanghai, China). Available potassium (AK) was extracted with 1 mol/L NH4OAc and measured using flame photometry. [41]. Total potassium (TK) was measured by NaOH melting and flame photometry [41]. Available nitrogen (AN) was measured with the alkali diffusion method [41].

### 2.4. DNA Extraction and PCR Amplification

Microbial DNA was extracted from the six samples using the MP-Fast DNATM Spin Kit for soil (Omega Bio-tek, Norcross, GA, USA) according to the manufacturer's protocols. The final DNA concentration and purification were determined using a NanoDrop 2000 UV-vis spectrophotometer (Thermo Scientific, Wilmington, NC, USA), and DNA quality was checked by 1% agarose gel electrophoresis.

### 2.5. PCR Amplification and Illumina MiSeq Sequencing

The V3–V5 hypervariable regions of the bacterial 16S rRNA gene were amplified with primers 515F (5′-GTGCCAGCMGCCGCGG-3′) and 907R (5′-CCGTCAATTCMTTTRAGTTT-3′). The fungal 18S rRNA gene was amplified with primers SSU0817F (5′-TTAGCATGGAATAATRRAATAGGA-3′) and 1196R (5′-TCTGGACCTGGTGAGTTTCC-3′) using a thermocycler polymerase chain reaction (PCR) system (GeneAmp 9700, ABI, Carlsbad, CA, USA). The PCRs were conducted using the following program: 3 min of denaturation at 95 °C, 27/35 cycles of 30 s at 95 °C, 30 s of annealing at 55 °C, 45 s of elongation at 72 °C, and a final extension at 72 °C for 10 min. The PCRs were performed in triplicate using a 20-μL mixture containing 4 μL of 5 × FastPfu Buffer, 2 μL of 2.5 mM dNTPs, 0.8 μL of each primer (5 μM), 0.4 μL of FastPfu Polymerase, and 10 ng of template DNA. The resultant PCR products were extracted from a 2% agarose gel and further purified using the AxyPrep DNA Gel Extraction Kit (Axygen Biosciences, Union City, CA, USA) and then quantified using QuantiFluor-ST fluorometer (Promega, WI, USA) according to the manufacturer's protocol.

Purified amplicons were equimolar-pooled and paired-end sequenced (2 × 250 bp) on an Illumina MiSeq platform (Illumina, San Diego, CA, USA) according to the standard protocols by Majorbio Bio-Pharm Technology Co. Ltd. (Shanghai, China).

*2.6. Processing of Sequencing Data*

Raw FASTQ files were demultiplexed, quality filtered using the Trimmomatic tool, and merged using the FLASH tool with the following criteria: (i) The reads were truncated at any site receiving an average quality score of <20 over a 50-bp sliding window. (ii) Primers were exactly matched allowing two nucleotides to mismatch, and reads containing ambiguous bases were removed. (iii) Sequences with overlap longer than 10 bp were merged according to their overlap sequence.

*2.7. Statistical Analysis*

Operational taxonomic units (OTUs) were clustered with a 97% similarity cutoff using UPARSE software (version 7.1, http://drive5.com/uparse/, San Francisco, CA, USA), and chimeric sequences were identified and removed using UCHIME software (version 4.2, http://drive5.com/uchime/, San Francisco, CA, USA). All sequences were randomly sampled using Mothur software (version 1.39.5, https://github.com/mothur/, Detroit, MI, USA) to construct a Rarefaction curve based on the number of extracted sequences and the number of OTUs they could represent. The Chao index and Shannon index were calculated based on OTU dilution curve analysis. The taxonomy of each 16S rRNA and 18S rRNA gene sequence was analyzed using the RDP Classifier algorithm (http://rdp.cme.msu.edu/) against the Silva (SSU123) database using a confidence threshold of 70%. The community composition of each sample was calculated at each taxonomic level. The relationships between soil chemical indices and microbial (bacterial and fungal) communities were analyzed by using Redundancy analysis (RDA). All the data were analyzed on the free online platform of Majorbio I-Sanger Cloud Platform (http://www.i-sanger.com) (Shanghai Majorbio Bio-pharm Technology Co., Ltd., Shanghai, China). A one-way analysis of variation (ANOVA) and a least significant difference (LSD) multiple comparison test ($p < 0.05$), which were used to estimate variations in soil properties (soil nutrients and pH) among four forest management treatment methods. Differences were analyzed using SPSS software version 16.0 (SPSS Inc., Chicago, IL, USA).

**3. Results**

*3.1. Soil Properties*

Compared with the CK treatment, CNFM significantly decreased the soil AP concentration, while SFCS significantly increased the soil AP concentration (Table 2). Though the SOM, TN, total phosphorus (TP), TK, AN, and AK concentrations and pH value exhibited some differences between the three forest management treatments, the differences were not significant (Table 2).

**Table 2.** Soil chemical properties for the soil sample of forest under different management treatments.

| Modes | SOM (g kg$^{-1}$) | TN (g kg$^{-1}$) | TP (g kg$^{-1}$) | TK (g kg$^{-1}$) | AN (mg kg$^{-1}$) | AP (mg kg$^{-1}$) | AK (mg kg$^{-1}$) | pH |
|---|---|---|---|---|---|---|---|---|
| A | 60.57 ± 14.18 | 2.70 ± 0.72 | 0.40 ± 0.10 | 18.63 ± 1.30 | 205.22 ± 39.57 | 5.82 ± 0.72b | 133.62 ± 31.76 | 6.65 ± 0.15 |
| B | 61.28 ± 8.48 | 2.77 ± 0.31 | 0.40 ± 0.04 | 17.40 ± 1.97 | 207.76 ± 23.13 | 6.90 ± 1.78bc | 132.50 ± 54.11 | 6.20 ± 0.27 |
| C | 64.78 ± 11.75 | 2.86 ± 0.53 | 0.38 ± 0.06 | 18.55 ± 2.10 | 205.13 ± 33.38 | 9.61 ± 2.22c | 123.75 ± 32.32 | 5.99 ± 0.64 |
| D | 56.76 ± 17.02 | 2.49 ± 0.76 | 0.38 ± 0.10 | 20.72 ± 1.53 | 200.06 ± 53.97 | 13.46 ± 3.43a | 226.87 ± 123.95 | 6.24 ± 0.06 |
| F | 0.246 | 0.269 | 0.042 | 2.472 | 0.026 | 9.043 | 1.862 | 2.321 |
| P | 0.862 (NS) | 0.847 (NS) | 0.988 (NS) | 0.112 (NS) | 0.994 (NS) | 0.002 | 0.190 (NS) | 0.127 (NS) |

The values in the table are mean value (± SD, *n* = 4). Significant differences among different management treatments are indicated by different letters at the 0.05 level. NS indicates no significant; SOM indicates soil organic matter; TN indicates total nitrogen; TP indicates total phosphorus; TK indicates total potassium; AN indicates available nitrogen; AP indicates available phosphorus; AK indicates available potassium. A indicates close-to-nature forest management (CNFM), B indicates structure-based forest management (SBFM), C indicates unmanaged control (CK), and D indicates secondary forest comprehensive silviculture (SFCS).

### 3.2. Microbial Community Diversity and Composition Analysis

Analysis of OTUs determination at the 97% similarity level showed that the Shannon and Chao indices of soil bacteria and fungal were not significantly different among the different management methods ($p > 0.05$) (Table 3).

**Table 3.** Diversity indices for soil microbial community among different forest management methods.

| Plot | Bacteria | | Fungi | |
|------|------------|-----------------------|---------------|----------------------|
|      | **Shannon** | **Chao** | **Shannon** | **Chao** |
| A | $6.19 \pm 0.09a$ | $1966.60 \pm 91.57a$ | $3.11 \pm 0.24a$ | $230.21 \pm 8.51a$ |
| B | $6.11 \pm 0.07a$ | $1914.90 \pm 50.37a$ | $2.95 \pm 0.21a$ | $220.50 \pm 10.11a$ |
| C | $6.08 \pm 0.08a$ | $1910.80 \pm 103.69a$ | $2.92 \pm 0.21a$ | $213.59 \pm 21.69a$ |
| D | $6.03 \pm 0.17a$ | $1953.70 \pm 22.67a$ | $2.64 \pm 0.55a$ | $216.69 \pm 25.10a$ |

The values in the table are mean values ($\pm$ SD, $n = 4$). Significant differences among different management methods are indicated by different letters at the 0.05 level. A indicates close-to-nature forest management (CNFM), B indicates structure-based forest management (SBFM), C indicates unmanaged control (CK), and D indicates secondary forest comprehensive silviculture (SFCS).

The *Proteobacteria*, *Acidobacteria*, *Actinobacteria*, *Planctomycetes*, *Nitrospirae*, *Chloroflexi*, *Bacteroidetes*, and *Gemmatimonadetes* were the dominant phyla (relative abundance >1%) in 16 soil bacterial communities (Figure 1 and Table S2, Supplementary Materials). The relative abundance of *Proteobacteria* in the SBFM treatment ($36.86 \pm 0.61\%$) was significantly lower than that in the CK ($38.66 \pm 1.33\%$) treatment ($p = 0.049$), while the relative abundance of *Acidobacteria* in the SBFM treatment ($26.24 \pm 1.17\%$) was significantly higher than those in the CK treatment ($21.06 \pm 2.04\%$) ($p = 0.004$) and CNFM treatment ($21.35 \pm 1.53\%$) ($p = 0.002$). In addition, the relative abundance of *Bacteroidetes* in the CNFM treatment ($3.38 \pm 0.14\%$) was significantly higher than those in the CK treatment ($2.64 \pm 0.52\%$) ($p = 0.035$) and SBFM treatment ($2.76 \pm 0.29\%$) ($p = 0.009$).

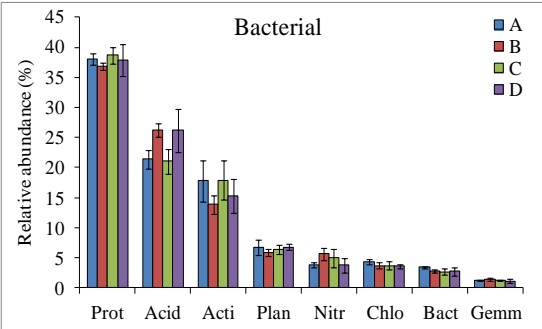 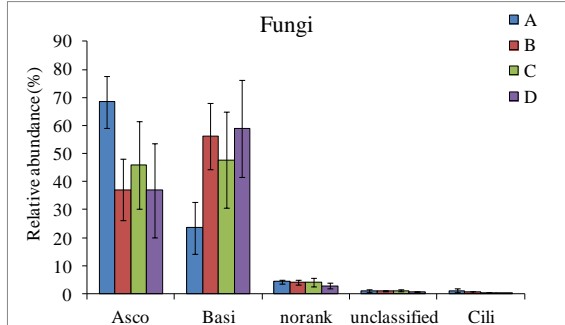

**Figure 1.** The composition and relative abundance of the soil microbial community at the phylum level. Prot indicates *Proteobacteria*, Acid indicates *Acidobacteria*, Acti indicates *Actinobacteria*, Plan indicates *Planctomycetes*, Nitr indicates *Nitrospirae*, Chlo indicates *Chloroflexi*, Bact indicates *Bacteroidetes*, Gemm indicates *Gemmatimonadetes*, Asco indicates Ascomycota, Basi indicates *Basidiomycota*, norank indicates *norank_k__Fungi*, unclassified indicates *unclassified_k__Fungi*, and Cili indicates *Ciliophora*. A indicates close-to-nature forest management (CNFM), B indicates structure-based forest management (SBFM), C indicates unmanaged control (CK), and D indicate secondary forest comprehensive silviculture (SFCS).

The dominant fungal phyla (relative abundance >1%) were *Ascomycota*, *Basidiomycota*, *norank_k__Fungi*, *unclassified_k__Fungi*, and *Ciliophora* (Figure 1 and Table S3, Supplementary Materials). The *Ascomycota* in the CNFM treatment ($68.46 \pm 9.29\%$) was significantly higher than those in the CK ($45.95 \pm 15.6\%$) ($p = 0.047$) and other treatments ($p = 0.004$; $p = 0.016$), while the *Basidiomycota* in the CNFM treatment ($23.60 \pm 9.23\%$) was significantly lower than those in the CK ($47.79 \pm 17.01\%$) ($p = 0.049$) and other treatments ($p = 0.005$; $p = 0.012$). In addition, the relative abundances of *Ciliophora*

in the CNFM (1.14 ± 0.65%) and SBFM (0.62 ± 0.30%) treatments was significantly higher than that in the SFCS treatment (0.21 ± 0.05%) ($p = 0.030$; $p = 0.034$).

The shared OTUs of the bacterial community in the four forest management treatments was 1875 phylotypes, and accounted for 86.60%, 86.65%, 87.45% and 87.45% of total phylotypes in CNFM, SBFM, CK, and SFCS treatment, respectively (Figure 2a). The majority of the shared OTUs were *Proteobacteria*, *Acidobacteria*, *Planctomycetes*, etc. (Figure S1a, Supplementary Materials); *Proteobacteria* and *Acidobacteria* were significantly different between the SBFM treatments and CK treatment. Moreover, shared OTUs of the fungal community in the four forest management treatments were 204 phylotypes, and accounted for 69.86%, 73.33%, 72.85%, and 79.37% of total phylotypes in the CNFM, SBFM, CK, and SFCS treatments, respectively (Figure 2b). The majority of the shared OTUs were *Ascomycota*, *Basidiomycota* and *norank_k__Fungi* (Figure S1b); *Ascomycota* and *Basidiomycota* were significantly different between the CNFM treatment and CK treatment.

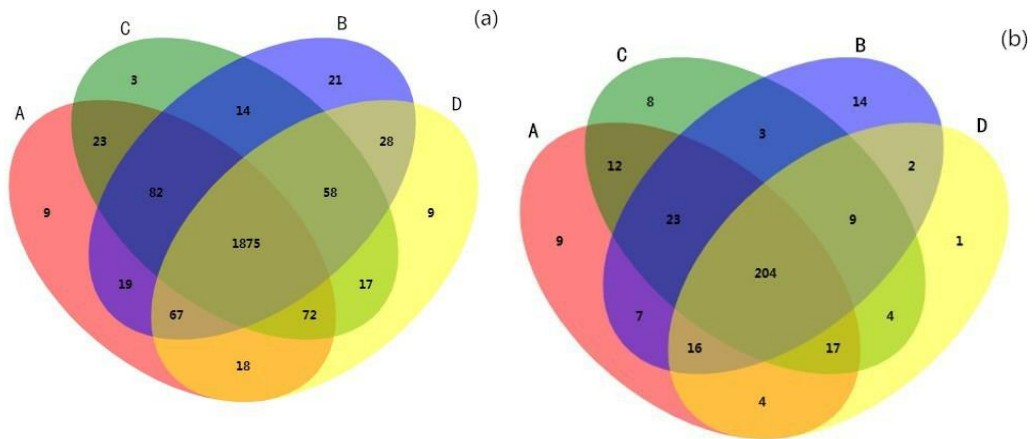

**Figure 2.** Venn diagrams based on OTUs of soil bacterial (**a**) and fungal (**b**) community with different forest management methods. A indicates close-to-nature forest management (CNFM), B indicates structure-based forest management (SBFM), C indicates unmanaged control (CK), and D indicate secondary forest comprehensive silviculture (SFCS).

Figure 3 shows the analysis result of Partial Least Squares Discriminant Analysis (PLS-DA), its results showed that the soil bacterial communities that were treated by CK and SFCS tend to be clustered together, and the SFCS and SBFM sited also tended to be clustered together, while CK sites were significantly separated from CNFM and SBFM treatments, and CNFM and SBFM were separated from each other. Meanwhile, the fungal communities that were treated by CK and SFCS tend to be clustered together, while the CK sites were significantly separated from CNFM and SBFM treatments, and CNFM and SBFM were separated from each other.

### 3.3. Correlations between Soil Properties and Microbial Communities

As shown in Table S3 (Supplementary Materials), the relative abundances of the bacterial phyla were not significantly correlated with soil nutrients ($p > 0.05$); however, the abundance of *norank_c__Acidobacteria* was positively correlated with SOM ($p = 0.02$), TN ($p = 0.02$), and AN ($p = 0.06$) (Table S5, Supplementary Materials). The abundance of *Rhodospirillales* belonging to the phylum *Proteobacteria* was positively correlated with TK ($p = 0.034$). The abundance of *Gaiellales* belonging to the phylum *Actinobacteria* was positively correlated with TP ($p = 0.004$) (Table S5, Supplementary Materials).

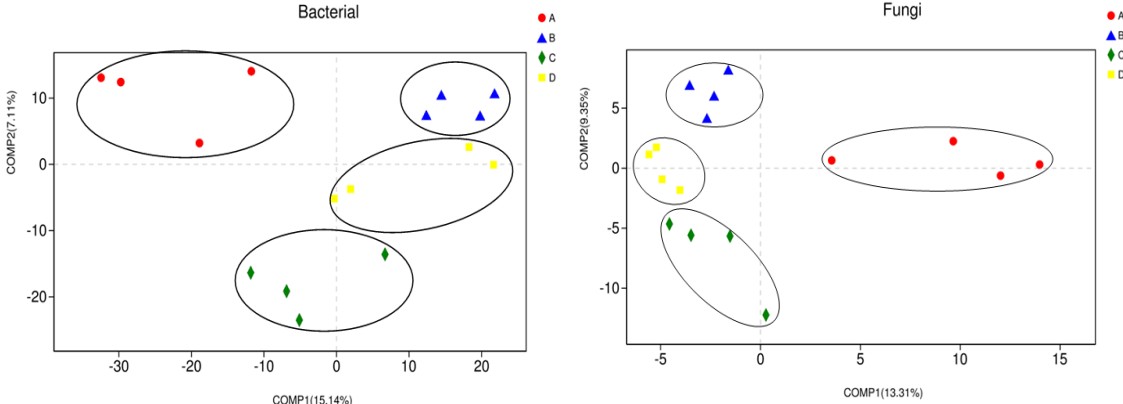

**Figure 3.** Structural similarity analysis of soil microbial communities among different management treatments by partial least squares discriminant (PLS-DA). A indicates close-to-nature forest management (CNFM), B indicates structure-based forest management (SBFM), C indicates unmanaged control (CK), and D indicates secondary forest comprehensive silviculture (SFCS).

Furthermore, the relative abundances of the fungal phyla, *Basidiomycota* and *Ciliophora* had significant negative correlations with pH ($p = 0.038$) and TK ($p = 0.041$) (Table S6, Supplementary Materials), and the abundance of *Sordariales* belonging to the phylum *Ascomycota* was positively correlated with SOM ($p = 0.027$), TN ($p = 0.036$), AN ($p = 0.024$), and TP ($p = 0.04$) (Table S7 Supplementary Materials).

### 3.4. Effect of Soil Properties on Microbial Communities at the Genus Level

Redundancy analysis (RDA) showed a clear association between soil environmental factors and microbial communities, which indicated that the RDA axes explained 37.56% and 59.14% of the total variations in the soil bacterial and fungal community compositions, respectively (Figure 4). The results showed that the contents of SOM, TN, and AN were correlated with the bacterial community composition at the genus level ($p < 0.05$), and the AK content was significantly correlated with the composition of the fungal communities at the genus level ($p < 0.05$) (Table 4).

**Table 4.** Correlation analysis between RDA and soil properties with soil microbial at the genus level.

| | Bacteria | | | | Fungi | | | |
|---|---|---|---|---|---|---|---|---|
| | **RDA1** | **RDA2** | $r^2$ | $p$ | **RDA1** | **RDA2** | $r^2$ | $p$ |
| SOM | 0.757 | −0.653 | 0.421 | 0.031 * | 0.947 | −0.320 | 0.094 | 0.529 |
| TN | 0.766 | −0.642 | 0.432 | 0.030 * | 0.907 | −0.421 | 0.131 | 0.383 |
| TP | 0.961 | 0.276 | 0.024 | 0.859 | 0.802 | 0.596 | 0.253 | 0.141 |
| TK | −0.901 | −0.432 | 0.102 | 0.449 | 0.978 | 0.205 | 0.008 | 0.942 |
| AN | 0.848 | −0.53 | 0.428 | 0.031 * | 0.954 | −0.299 | 0.204 | 0.207 |
| AP | −0.906 | −0.421 | 0.013 | 0.909 | 0.999 | 0.014 | 0.048 | 0.725 |
| AK | 0.892 | 0.451 | 0.114 | 0.472 | 0.996 | 0.086 | 0.452 | 0.035 * |
| pH | −0.767 | −0.641 | 0.221 | 0.200 | −0.914 | 0.403 | 0.135 | 0.397 |

The $r^2$ indicates the proportion of variance explained. ** ($p < 0.01$) and * ($p < 0.05$) indicate significant differences among values of soil sample parameters based on a one-way ANOVA followed by an least significant difference (LSD) test. SOM indicates soil organic matter; TN indicates total nitrogen; TP indicates total phosphorus; TK indicates total potassium; AN indicates available nitrogen; AP indicates available phosphorus; AK indicates available potassium.

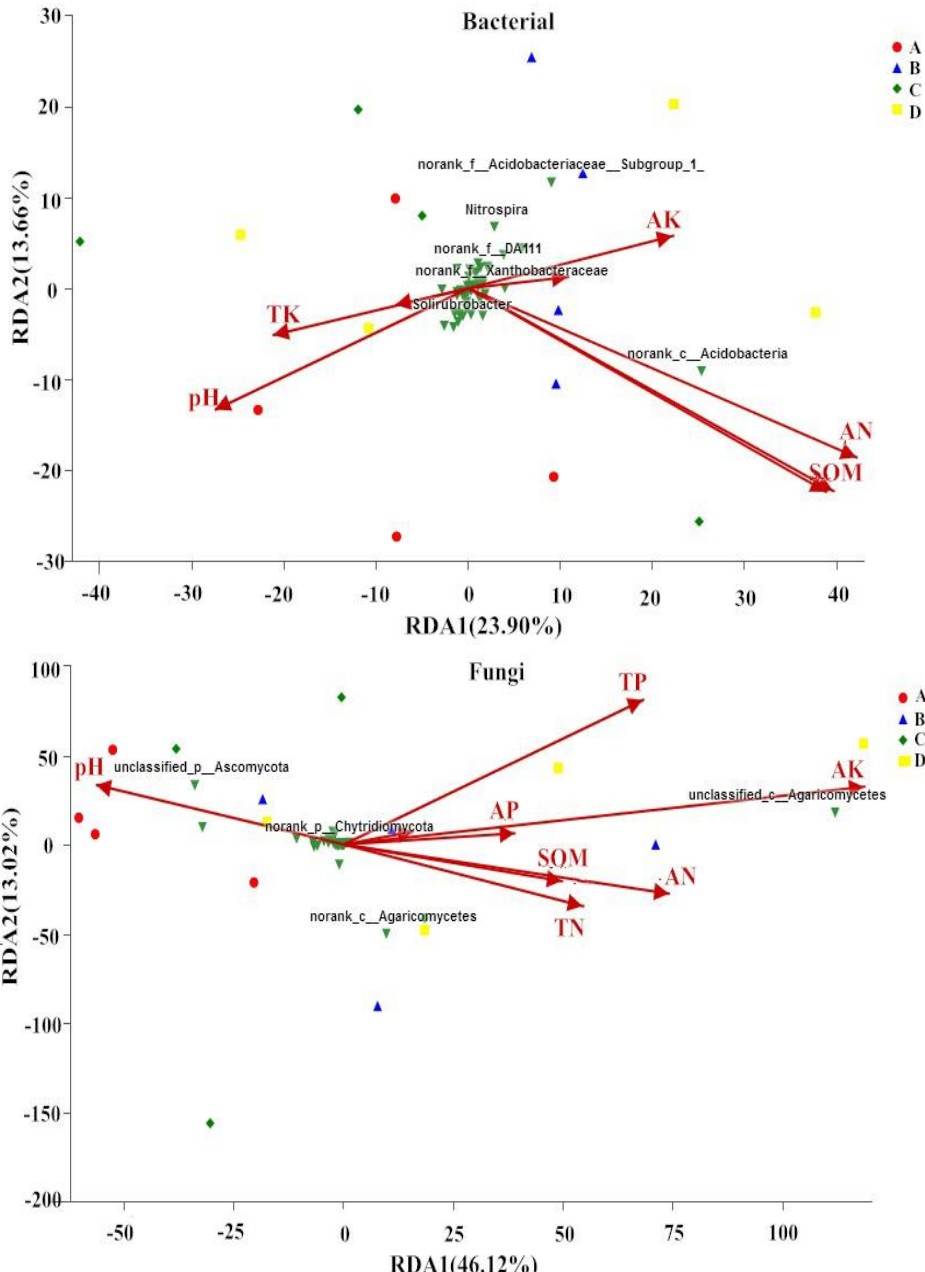

**Figure 4.** Redundancy analysis (RDA) of the relationships between soil physiochemical properties and bacterial and fungi communities under different forest management methods. A indicates close-to-nature forest management (CNFM), B indicates structure-based forest management (SBFM), C indicates unmanaged control (CK), and D indicate secondary forest comprehensive silviculture (SFCS).

## 4. Discussion

### 4.1. Forest Management Effects on Soil Nutrients

Our study indicated that the contents of soil nutrients other than AP were not significantly different among the four treatments, which showed that soil nutrients did not change in the mixed oak forest after tending in the short term (Table 2); the reason for that result may be that the forest management effect has not been fully demonstrated because the management time was the short term [42,43].

The AP concentration was observed in the SFCS treatment soil samples than in CK treatment soil samples (Table 2), and the results indicated that SFCS could improve the effectiveness of Phosphorus. In addition, we found that the soil pH value was not significantly different among the different management treatments in the short term (Table 2), and the result was similar to the observation by Yang et al. [44] and Dang et al. [12], who also showed that forest management caused no obvious change in the pH value of forest soil.

### 4.2. Changes in Soil Microbial Community Composition and Diversity

In the forest ecosystem, trees can change the forest microclimate, and they can produce exudation from roots, litter, and wood debris; meanwhile, trees interact with soil microbes through roots, and thus can influence ecosystem properties [14]. Hence, the composition of soil microbial communities in forest soil could be determined by the dominant trees of the forest stand [15]. However, our study showed that the bacterial and fungal community diversity and richness were not significantly different among the four treatments (Table 3). This result was consistent with the results of Dang et al. [12], who showed that thinning had no significant effect on the diversity of soil microbial communities. The reason may be that the main tree species and litter properties type did not change during the different forest management methods in a short time.

While forest management can change the dominant phyla of the microbial communities (Figure 1), previous researches found that *Proteobacteria*, the most bacterial phyla, were related to the soil carbon and nitrogen cycling [17,45]. Hence, Proteobacteria are commonly regarded as copiotrophic bacteria [46,47]. Conversely, *Acidobacteria* belong to oligotrophic bacteria [48,49]. In our study, the relative abundance of *Proteobacteria* was higher than that of *Acidobacteria* in all treatments (Table S2, Supplementary Materials), which indicated that the soil nutrient condition in this forest stand was copiotrophic to some extent. Moreover, our study also showed that the relative abundance of Proteobacteria was significantly lower in the SBFM compared with in the CK treatment (Table S2, Supplementary Materials), while the relative abundance of *Acidobacteria* was significantly higher in the SBFM compared with those in the CK treatment and CNFM treatment (Table S2, Supplementary Materials). These results predicted that the soil nutrient condition of forest with SBFM treatment may cause higher oligotrophic soil compared with that in the CK treatment. In addition, the relative abundance of Bacteroidetes was significantly higher in the CNFM compared with that in the CK treatment, and other studies showed that Bacteroidetes are also considered to be copiotrophic bacterial phyla [46–48]; hence, forest soil subjected to CNFM treatment may cause higher copiotrophic soil compared with that in the CK treatment. However, the relative abundance of bacterial phyla had no significant difference in CK and SFCS treatments (Table S2, Supplementary Materials), which indicated that SFCS did not alter the bacterial community. Unfortunately, the level of soil nutrients could not be well reflected by the copiotrophic- or oligotrophic-dominant bacterial phyla because soil nutrients did not significantly differ in the forest management treatments.

The soil fungal community was dominated by *Ascomycota* and *Basidiomycota*, and their changes were driven by the presence of the phyla *Ascomycota* and *Basidiomycota*. Fungi are heterotrophic organisms; their survival mainly depends on exogenous carbon that comes from the rhizosphere and soil. This exogenous carbon mainly includes primary metabolites and secondary metabolites that are produced by plant roots [50,51]. However, the roots of different plants produce different types of exogenous carbon, which may determine the composition of fungal communities. Tikkanen et al. [52] showed that fungi associated with symbiotic root systems depend on plant species. Our study showed that the relative abundance of Ascomycota was significantly higher in the CNFM treatment compared with that in the CK treatment and others treatments (Table S3, Supplementary Materials), while the relative abundance of *Basidiomycota* was significantly lower in the CNFM treatment compared with those in the CK treatment and CNFM treatment (Table S3, Supplementary Materials), in agreement with the study of Hartmann et al. [53], who showed that harvesting had effects on the fungal communities, namely, the abundance of *Ascomycota* was increased, and that of *Basidiomycota* was decreased in

harvested stands. Similarly, a study showed that the abundance of ectomycorrhizal species belonging to the Basidiomycota was consistently lower in harvested plots compared with that in unharvested plots [53]. Hence, these results indicated that CNFM can significantly reduce the amount of tree litter and roots, which may lead to the decrease of ectomycorrhizal belonging to the phylum Basidiomycota and increase of the abundance of Ascomycota after four years of tending with CNFM.

### 4.3. The Relationship between Soil Nutrients and Microbial Communities

Many studies have reported that soil pH and soil nutrients might influence soil bacterial communities. This study shows that the dominance of bacterial phyla was not significantly correlated with soil nutrients (Table S4, Supplementary Materials), which is not consistent with other studies [12,48,54], suggesting that these bacterial phyla were not sensitive to the soil nutrient levels. Interestingly, we found that the dominance of the bacterial order had a significant correlation with soil nutrients, such as the SOM, TN, and AN, which correlated positively with *norank_c__Acidobacteria*. The TK correlated positively with *Rhodospirillales*, and the TP correlated positively with *Gaiellales* (Table S5, Supplementary Materials). Moreover, *Rhodospirillales* and *Gaiellales* belonged to the phylum *Proteobacteria* and *Actinobacteria* respectively. Among these results, *Gaiellales* positively correlated with TP, which is consistent with the report by Liu et al. [54], who reported that *Actinobacteria* positively correlated with TP. Furthermore, Albertsen et al. [55] showed that *Actinobacteria* participate in the biological phosphorus removal process. In addition, previous studies showed that the soil pH value did not correlate with the dominance of bacterial communities [17,56]. Our study showed the same results.

The copiotroph and oligotroph are commonly used to indicate the relationship between microbial and environmental ecological properties [48]. Copiotrophs are found in environments which are rich in nutrients, particularly carbon; however, opposite to copiotrophs, oligotrophs survive in much lower carbon concentrations [48]. Generally, *Proteobacteria* and *Bacteroidetes* are considered to be copiotrophs since they survive in much higher organic carbon concentrations, and *Acidobacteria* are oligotrophs because they prefer nutrient-poor environments. In our study, soil nutrient status was not well reflected by the changes of the abundance of *Proteobacteria* and *Acidobacteria*, because the soil nutrients did not significantly correlate with the abundance of copitrophic Proteobacteria and oligotrophic *Acidobacteria*.

Fungi prefer acid soil environments [57,58], and its diversity indices had a significantly negative association with pH value [57]. Hence, soil pH can influence and determine soil fungal communities [59]. In this study, we found that the abundances of the phyla Basidiomycota had significant negative correlations with pH value (Table S6, Supplementary Materials), similar to the observation by Wang et al. [60], who showed that the relative abundance of the phylum *Basidiomycota* negatively correlated with pH. In addition, Wang et al. [60] reported that *Ascomycota* positively correlated with TN, soil organic carbon and Liu et al. [54] also reported that the abundance of *Ascomycota* correlated with the AP content. We found that the abundances of the order *Sordariales* belonging to the phylum *Ascomycota* had significant positive correlations with SOM, TN, AN, and TP (Table S7, Supplementary Materials). The results were similar to the previous studies, suggesting that the changes in the *Sordariales* abundances were due to the P level.

The SOM, TN, and AN correlated with the bacterial communities, while the AK was the only soil nutrient that correlated with the fungal communities (Table 4), indicating that the soil nutrients could significantly affect bacterial communities, but had little effect on soil fungal communities. The bacterial communities mainly inhabit soil and are not directly related to plant roots or the rhizosphere, which are mainly affected by soil abiotic factors [61]. Meanwhile, bacteria can utilize simple organic decomposable substrates and compete for nutrients in other microbial groups, and can maintain a relatively stable community composition and structure [62]. Unlike the bacterial community, the fungal communities form plant-fungal symbiosis with aboveground plants [63], so that fungi are often related to specific plants and can be used as lignin decomposing or litter decomposition agents [15]. Hence, the soil nutrients had greater effects on the bacterial community than the fungal community, and the soil fungal community may be primarily affected by its host plants.

## 5. Conclusions

The purpose of this research was to understand the effect of different forest management methods on soil nutrients and microbial communities. The results illustrated that the forest management did not change the levels of soil nutrients other than AP in the short term. Although we observed little effect on the diversity of soil microbial communities, we see significant shifts in dominant microbial communities among treatments. The SBFM altered microbial bacterial community compositions, and CNFM altered microbial fungal community compositions. SFCS did not alter the bacterial and fungal community compositions. For bacterial, *Proteobacteria* was lower in the SBFM treatment, and *Acidobacteria* was higher in the SBFM treatment. However, the soil nutrients did not significantly correlate with the abundance of copitrophic *Proteobacteria* and oligotrophic *Acidobacteria*, which means that the soil nutrient status was not well reflected by the changes of the abundances of *Proteobacteria* and *Acidobacteria*. For fungal, *Ascomycota* was more abundant in the CNFM treatment, and *Basidiomycota* was lower in the CNFM treatment. In addition, our results also indicated that the soil nutrients had effects on microbial communities, and their influence was greater on the bacterial community than on the fungal community.

**Supplementary Materials:** The following are available online at http://www.mdpi.com/1999-4907/10/2/161/s1, Figure S1: Core microbial community composition of bacterial (**a**) and fungal (**b**) in four treatments, Table S1: Differences among the three forest management methods, Table S2: Composition of bacterial community at the phylum level, Table S3: Composition of fungi community at the phylum level, Table S4: Correlation between soil nutrients and relative abundance of the dominant phyla in the bacterial community, Table S5: Correlation between soil nutrients and relative abundance of the dominant order in the bacterial community, Table S6: Correlation between soil nutrients and relative abundance of the dominant phyla in the fungal community, Table S7: Correlation between soil nutrients and relative abundance of the dominant order in the fungal community.

**Author Contributions:** P.W. jointly conceived the study with G.H.; G.Z., W.L., Z.Z., and P.W. designed the experiments and collected data; P.W., G.Z., and G.H. presented the method; Y.H., P.W., and G.Z. analyzed the data; P.W. wrote the manuscript. All authors discussed the results and reviewed the manuscript.

**Funding:** This research was funded by finical support of National Key Research and Development Program of China (grant number: 2016YFD0600203).

**Conflicts of Interest:** The authors declare no conflict of interest.

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
