# Peer review of "Short-Term Effects of Different Forest Management Methods on Soil Microbial Communities of a Natural Quercus aliena var. acuteserrata Forest in Xiaolongshan, China"

_forests, doi:10.3390/f10020161_

Round 1
Reviewer 1 Report
The manuscript is very interesting and has been properly prepared. Only, English language requires a small correction.
Author Response
Response: Thanks for your helpful comments. The English in this document has been checked by MDPI Language Editing Services, and we read the manuscript carefully after the English revision to convince that the text was revised correctly.
Reviewer 2 Report
Manuscript ID forests-438337 tilted “Short-term effects of different forest management methods on soil microbial communities of a Natural Quercus aliena var. acuteserrata Forest in Xiaolongshan, China” is showing interesting outcomes for the scientific community. Results are good and discussed properly and few minor corrections will improve the results representation. The manuscript needed a language expert to reduce the language and grammatical errors. Revised the manuscript as per the points below.
1) English language needs improvement.
2) Fig. 1 and Fig.2 should be reconstructed based on individual bacterial and fungal phylum that represents the clear shift of microbial community and comparison among different forest management methods.
3) A venn diagram based on OTUs should be give an idea of core-microbial community and their shift in between different forest management methods.
Author Response
-- 1. English language needs improvement.
Response: Thanks for your careful reading of our manuscript. The English in this document has been checked by MDPI Language Editing Services, and we read the manuscript carefully after the English revision to convince that the text was revised correctly.
-- 2. Fig. 1 and Fig. 2 should be reconstructed based on individual bacterial and fungal phylum that represents the clear shift of microbial community and comparison among different forest management methods.
Response: Thanks for your helpful comments. We agree with you and have reconstructed the Fig. 1 and Fig.2, which represents the clear shift of microbial community and comparison among different forest management methods in my revised manuscript. We hope that you agree.
-- 3. A venn diagram based on OTUs should be give an idea of core-microbial community and their shift in between different forest management methods.
Response: Thanks for your helpful comments. Following your comments, we have added a venn diagram in my revised manuscript. We hope that you agree.
Reviewer 3 Report
Manuscript Tilted “Short-term effects of different forest management methods on soil microbial communities of a Natural Quercus aliena var. acuteserrata Forest in Xiaolongshan, China” is scientifically sound and outcome of the research is good to publish. Results are well explained in discussion section. Writing part is weak there are many spelling and grammar mistakes, although manuscript is will written but still it needs to be proof read once again. Plagiarism is also an issue few sentences are copied and pasted without paraphrasing. Plagiarism report is attached here.
Below are some specific point, should be dealt by the author
71 should be “could” instead “can”
81 should be “community. ” instead “community ”
83 should be “management ” instead “managements ”
93 should be “ silviculture” instead “silvicuture ”
100 should be “ silviculture” instead “silvicuture ”
103 should be “management ” instead “managements ”
107 should be “management ” instead “managements should be “ ” instead “ ”
136 should be “ conditions” instead “condition ”
138 should be “ silviculture” instead “silvicuture ”
142 should be “unmanaged ” instead “unmanged ”
148 should be “ collect” instead “collected ”
161 should be “ colourimetric” instead “colorimetric ”
219 should be “ silviculture” instead “silvicuture ”
226 statement is not clear “The Shannon and Chao indices of soil bacteria had little difference among
different managements, but the difference was not significant (P>0.05) (Table 3) ”
229 do you want to write “ little” instead “ litter”
257 should be “no ” instead “not”
260, 270, 276, 312 should be “ silviculture” instead “silvicuture ”
328 should be “ could” instead “can ”
347 should be “ is” instead “are ”
357 should be “its being associated ” instead “it beingassociated ”
387 should be “ might” instead “may ”
399 should be “ dominance” instead “dominace ”
425 should be “ community” instead “communitiy ”
433 should be “ primarily” instead “primary ”
436 should be “ change” instead “changes ”
443 should be “ significantly ” instead “siginificant ”
451 should be “Acknowledgement ” instead “Acknowledgments ”

Author Response
-- 1. L71. should be “could” instead “can” ; L81. should be “community. ” instead “community ”; L83. should be “management ” instead “managements ”; L93. should be “ silviculture” instead “silvicuture ”; L100. should be “ silviculture” instead “silvicuture ”; L103. should be “management ” instead “managements ”; L107. should be “management ” instead “managements should be “ ” instead “ ” ; L136. should be “ conditions” instead “condition ” ; L138. should be “ silviculture” instead “silvicuture ” ; L142. should be “unmanaged ” instead “unmanged ”; L148. should be “ collect” instead “collected ”; L161. should be “ colourimetric” instead “colorimetric ”; L219. should be “ silviculture” instead “silvicuture ”
Response: Thanks for your helpful comments. I am sorry for these mistakes, we have corrected these mistakes according your comments in my revised manuscript. We hope that you agree.
-- 2. L226. statement is not clear “The Shannon and Chao indices of soil bacteria had little difference among different managements, but the difference was not significant (P>0.05) (Table 3) ”
Response: Thanks for your helpful comments. I am sorry for the mistakes. This sentence that is the Shannon and Chao indices of soil bacteria was not significant difference among different managements. So, in order to express clearly, we revised this sentence in my revised manuscript. As follow: The Shannon and Chao indices of soil bacteria was not significantly differenct among different management methods (P>0.05) (Table 3). We hope that you agree.
-- 3. L 229 do you want to write “ little” instead “ litter”
Response: Thanks for your helpful comments. Yes, we want to write “ little” instead “ litter”, and we have corrected this word in my revised manuscript. I am sorry for these careless mistakes again.
-- 4. L257. should be “no ” instead “not” ; L260, L270, L276, L312 should be “ silviculture” instead “silvicuture ”; L328. should be “ could” instead “can ” L.347 should be “ is” instead “are ”
L357. should be “its being associated ” instead “it being associated ”; L387. should be “ might” instead “may” ; L399. should be “ dominance” instead “dominace ”; L425. should be “ community” instead “communitiy ”; L433. should be “ primarily” instead “primary ”; L436. should be “ change” instead “changes ” L.443 should be “ significantly ” instead “siginificant ”; L451. should be “Acknowledgement ” instead “Acknowledgments ”
Response: Thanks for your helpful comments. We have corrected these mistakes in my revised manuscript.